# Hub-Pathway: Transfer Learning from A Hub of Pre-trained Models

**Yang Shu, Zhangjie Cao, Ziyang Zhang[§], Jianmin Wang, Mingsheng Long[✉]**
School of Software, BNRist, Tsinghua University, China
[§]Advanced Computing and Storage Lab, Huawei Technologies Co. Ltd
{shuyang5656,caozhangjie14}@gmail.com, {jimwang,mingsheng}@tsinghua.edu.cn

## Abstract

Transfer learning aims to leverage knowledge from pre-trained models to benefit the target task. Prior transfer learning work mainly transfers from a single model. However, with the emergence of deep models pre-trained from different resources, model hubs consisting of diverse models with various architectures, pre-trained datasets and learning paradigms are available. Directly applying single-model transfer learning methods to each model wastes the abundant knowledge of the model hub and suffers from high computational cost. In this paper, we propose a *Hub-Pathway* framework to enable knowledge transfer from a model hub. The framework generates data-dependent pathway weights, based on which we assign the pathway routes at the input level to decide which pre-trained models are activated and passed through, and then set the pathway aggregation at the output level to aggregate the knowledge from different models to make predictions. The proposed framework can be trained end-to-end with the target task-specific loss, where it learns to explore better pathway configurations and exploit the knowledge in pre-trained models for each target datum. We utilize a noisy pathway generator and design an exploration loss to further *explore* different pathways throughout the model hub. To fully *exploit* the knowledge in pre-trained models, each model is further trained by specific data that activate it, which ensures its performance and enhances knowledge transfer. Experiment results on computer vision and reinforcement learning tasks demonstrate that the proposed Hub-Pathway framework achieves the state-of-the-art performance for model hub transfer learning.

## 1 Introduction

With the emergence of large-scale datasets, various large-scale deep models are developed in wide range of areas such as computer vision [18, 65, 72] and natural language processing [16, 6]. These models not only show state-of-the-art performance in the tasks that they are designed for, but also achieve high performance when fine-tuned on various downstream tasks with different distributions. On the other hand, due to the requirement of large-scale annotated data, deep learning is still difficult to apply in specific domains without sufficient data [34]. Therefore, transfer learning, which aims to transfer knowledge from existing models and datasets to the target task, has attracted more and more attention recently [75, 6].

Current deep learning paradigm tends to increase the scale of the pre-trained model to embed more knowledge into the model, which can be transferred to more diverse tasks [25, 6]. However, infinitely increasing the model scale requires machines with larger storage and stronger GPUs, which are expensive and cost much energy. Instead, many pre-trained models from different resources are already available, including pre-trained on diverse datasets or tasks, e.g. ImageNet for classification [55], or BERT for language modelling [16], pre-trained with diverse learning paradigms, e.g. supervised learning [29], self-supervised learning [19] or unsupervised learning [9] and with various

architectures, e.g. single-stage [42] or two-stage models [28] for object detection. These diverse models contain abundant knowledge and introduce low costs in both training and inference. Thus, there is a high motivation to enable knowledge transfer from all of these models. We define the problem as *model hub transfer learning*, where a model hub consists of pre-trained models from different resources.

Human beings also face the same problem of transfer learning from a model hub, where the different pre-trained models are similar to different regions of our brains specializing in different tasks. When solving a new task, human beings only call on the most relevant region of the brain to address it. Recently, Dean et al. [15] also proposed the outlook that the next-generation AI architecture should activate different parts of the model to solve different tasks. Motivated by these, we design pathways to activate different pre-trained models for prediction depending on the input data. However, there exist two challenges in designing such pathways. Firstly, how to explore and find the optimal pathway configuration to activate the most relevant models for each target datum? Secondly, even knowing which pre-trained models to use, how to aggregate and exploit the knowledge in them?

We propose a new *Hub-Pathway* framework to address the above challenges, which enables effective knowledge transfer from a model hub. We design an architecture with the pathway route at the input level to select which pre-trained models to activate, and the pathway aggregation at the output level to aggregate the prediction from different pre-trained models. Both the pathway route and the pathway aggregation are based on pathway weights generated data-dependently by a network. We only activate the pathways with top $k$ weights to avoid negative transfer of irrelevant models and improve the efficiency of the framework. The framework is optimized by a task-specific loss to ensure that the pathways introduce high transfer performance. To promote the exploration of the pathways, we employ a noisy pathway generator and design an exploration loss to encourage the framework to explore and compare different pathways and find better pathway configurations. To promote the exploitation of the knowledge in the activated models, we further fine-tune the models with the specific data activating them. In all, our contribution can be summarized as:

- We investigate the important problem of transfer learning from a model hub and propose a general Hub-Pathway framework that can address different situations where models may be trained from different datasets and learning paradigms and with diverse architectures. We design data-dependent pathways to route different data to different models at the input level and aggregate the transferred knowledge at the output level to make predictions.

- We promote the exploration and exploitation of the framework to enhance more effective transfer learning from a model hub. We design a noisy pathway generator and an exploration loss to encourage exploring more models and pathway configurations. We further propose to fine-tune the activated models with specific data to fully utilize the transferred knowledge.

## 2   Related Work

**Transfer Learning.** Transfer learning is a learning paradigm to improve the performance of the target task by transferring knowledge from source domains [49]. Early works transfer knowledge by directly reusing features extracted by the pre-trained model [48, 17] and later works find that fine-tuning the pre-trained model introduces higher transfer performance [2, 23]. As an easy and effective transfer learning method, many empirical studies investigate the effectiveness of fine-tuning on various downstream tasks and scenarios, such as classification [37, 73], few-shot learning [53], medical imaging [52], object detection and instance segmentation [27]. Recently, several works have explored the limitations of the vanilla fine-tuning method and developed new techniques to improve the performance. To enable efficient transfer learning, residual adaptors [54] are introduced to tune only a few parameters for the target tasks. Parameters of fine-tuned models can be averaged, which promotes the performance while keeping the inference efficiency [64]. Various fine-tuning methods have also been proposed, including regularizing the parameters [67] or features [40, 10] and exploring category relationships [70] or category intrinsic structures [74] by auxiliary tasks. Several works study the process of transfer learning, such as transferability of each layer in a pre-trained model [69] and the effect of intrinsic dimensions [1]. However, these studies focus on fine-tuning from a single model, and it remains unclear how to extend single-model transfer learning techniques to multiple pre-trained models, where simply assembling every single model is inefficient.

**Transfer Learning with Multiple Models.** In the presence of multiple pre-trained models, recent works estimate the transferability of each pre-trained model and select the best one without the need of fully fine-tuning each of them [62, 3, 47, 71, 5, 33]. But subject to the inaccurate measurement of model transferability, which is a hard problem itself, such model-level selection may degrade the performance and lose the rich knowledge in all the models since only a single best model is utilized. To address this problem, several works propose to aggregate the features [56, 41] or parameters [59] of all the pre-trained models. However, all of these methods assume that the pre-trained models form a model zoo where all models have layer-to-layer correspondence or even share the same architecture, which is unrealistic for the most general case of transfer learning with arbitrary models. We instead propose model hub transfer learning to remove all these constraints on the pre-trained models and realize the most general case of transfer learning.

**Conditional Computation.** Our method is also related to conditional computation or dynamic neural networks [24], where parts of the network are active on a per-example basis. A type of conditional computation method employs conditionally parameterized networks, which produce an adaptive ensemble of the model parameters [68, 11]. A series of works [35, 14, 20, 13] introduce the idea of multiple mixture-of-experts (MoE) [46] activated by data-adaptive gating networks. This idea has been extended to foundation models, which employ mixture-of-experts layers to form outrageously large neural networks [58, 21]. Different from these methods which create mixture-of-experts from homogeneous networks with random initializations, our hub-pathway method simultaneously explores the pathways through different pre-trained models with diverse knowledge and exploits the relevant part of knowledge to enhance transfer learning performance on the downstream tasks.

## 3   Hub-Pathway

We first define the problem of model hub transfer learning. Then we introduce the Hub-Pathway framework with delicate designs to boost its exploration and exploitation of the model hub, simultaneously improving positive transfer of relevant knowledge and avoiding negative transfer.

The most common transfer learning scenario considers only one single pre-trained model $\Theta$ at hand to serve the target task $\mathcal{D} = \{(\mathbf{x}, \mathbf{y})\}$. In *model hub transfer learning*, we consider a situation where we have a hub of pre-trained models $\{\Theta_1, \Theta_2, \cdots, \Theta_m\}$, where all models can differ in all the aspects including the pre-training datasets, pretext tasks, learning paradigms and the network architectures. This problem setting can be generally applied in various realistic problems even if the pre-trained models are obtained from quite different resources.

Motivated by cognitive science and the pathway idea in deep networks, we propose a Hub-Pathway framework to address the problem. We illustrate the architecture of our framework in Figure 1. We modify pre-trained models by replacing their output head layers with target-task-specific output layers, which form the models we will transfer from. As our motivation stated above, different data have different relations with the pre-trained models and should activate and transfer knowledge from different models. To realize this idea, we employ a pathway generator network $G$, which outputs data-dependent pathway weights $G(\mathbf{x})$ based on the input data $\mathbf{x}$, with each dimension indicating the weight of one specific model in the hub. The dense pathway weights in $G(\mathbf{x})$ still activate all the pathways to all the pre-trained models. To utilize the most suitable pre-trained models for the target data, we only keep the top $k$ pathway weights and set the rest pathway weights as 0:

$$\bar{G}(\mathbf{x}) = f_{\text{topk}}\left(G(\mathbf{x}), k\right), \text{where } f_{\text{topk}}\left(G(\mathbf{x}), k\right)_i = \begin{cases} G(\mathbf{x})_i & G(\mathbf{x})_i \text{ in the top } k \text{ entries of } G(\mathbf{x}) \\ 0 & \text{otherwise.} \end{cases}$$

(1)

Such a design has two benefits: (1) the models with useful knowledge to each target datum are transferred but those of negative influence will be eliminated; (2) there are only data forwarding and back-propagation through the activated models, which makes both training and testing more efficient.

Based on the pathway weight $G(\mathbf{x})$, we assign the pathway route at the *input level* as $\mathbb{I}\left[\bar{G}(\mathbf{x}) > 0\right]$, which only passes the input data through the pre-trained models with a pathway weight large than 0. The pathways with a 0 pathway weight are blocked and the corresponding models are not used for the current data. Then at the *output level*, an aggregator network $A$ composes the outputs from different models with the pathway weights and outputs the final prediction of the model hub as $A\left(\left[\bar{G}(\mathbf{x})_i \cdot \Theta_i(\mathbf{x})\right]_{i=1}^m\right)$, where the function $[\cdot]$ concatenates all the $m$ vectors. To train the framework,

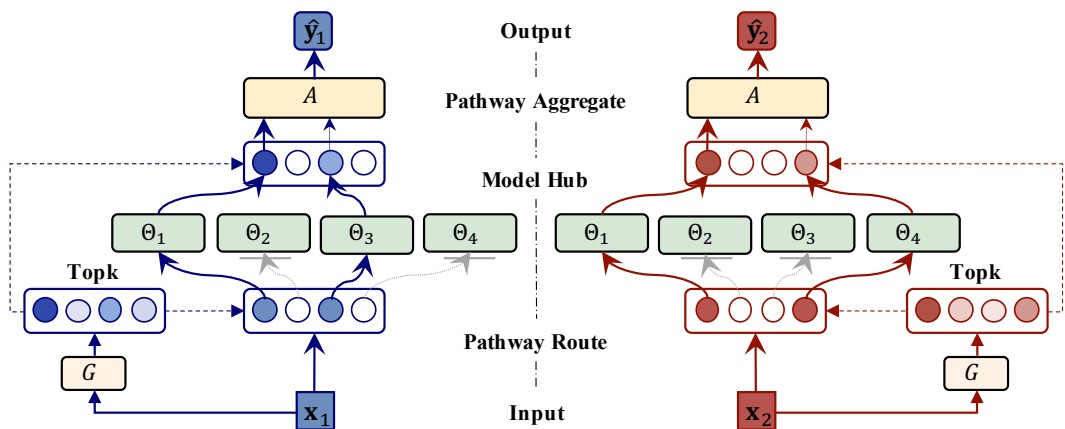

Figure 1: The architecture of our Hub-Pathway framework. We showcase using four pre-trained models $\Theta_1$, $\Theta_2$, $\Theta_3$, and $\Theta_4$, which are not changed in architecture but only fine-tuned in parameters. We generate the pathway weights by a generator network $G$ and only keep the top $k$ pathway weights while setting others as $0$. We set a pathway route at the *input level* and a pathway aggregation at the *output level* based on the pathway weights. The pathway route only makes the data pass the pathway with the weight large than $0$, while the pathway aggregation combines the predictions from different pre-trained models. A final aggregation network $A$ generates the final prediction based on the transferred knowledge. The different colors are to highlight that *for different input data, our framework introduces different pathways to make the best use of pre-trained knowledge*.

we then adopt a target-task-specific loss on the final prediction:

$$\mathcal{L}_{\text{task}} = \mathbb{E}_{(\mathbf{x},\mathbf{y})\sim\mathcal{D}}\ell\left(A\left(\left[\bar{G}(\mathbf{x})_i \cdot \Theta_i(\mathbf{x})\right]_{i=1}^{m}\right), \mathbf{y}\right), \tag{2}$$

where $\ell$ is the target-task loss, e.g. cross-entropy loss for the classification task. $\mathcal{L}_{\text{task}}$ ensures that the derived pathway configuration and model parameters contribute to the high performance on the target task, leading the framework to explore better pathways and exploit deeper the activated knowledge.

### 3.1 Enhancing Exploration on Hub Pathways

When optimized with only the task-specific loss, the pathway framework has the risk of over-fitting and falling to a local optimum by trivially activating and repeatedly updating with a few specific models, which ignores other potentially more useful models and wastes the rich knowledge in the hub. Such a phenomenon is called *hub collapse* in this paper. With the aim of encouraging the exploration of models with high transferability throughout the model hub, we design from the architecture level and the loss level by employing a noisy pathway generator and introducing an exploration loss.

In order to implement a noisy pathway generator, $G(\mathbf{x})$ is designed to embody a standard subnetwork $G_p$ and a randomized subnetwork $G_n$ with $\epsilon$ drawn from a standard Gaussian distribution:

$$G(\mathbf{x}) = \text{Softmax}\left(G_p(\mathbf{x}) + \epsilon \cdot \text{Softplus}(G_n(\mathbf{x}))\right), \epsilon \sim \mathcal{N}(0, 1), \tag{3}$$

where $\text{Softmax}(\mathbf{z}) = \frac{\exp(\mathbf{z})}{\sum_{z\in\mathbf{z}}\exp(z)}$ and $\text{Softplus}(\mathbf{z}) = \log(1+\exp(\mathbf{z}))$. The standard part $G_p(\mathbf{x})$ reflects which models are preferred and the noise term $G_n(\mathbf{x})$ adds some randomness to the pathway weights, which encourages the framework to explore other models or configurations during training.

Taking the key idea of maximum-entropy reinforcement learning that maximizes the exploration ability of the agent through the maximum entropy of the policy, we define the exploration loss $\mathcal{L}_{\text{explore}}$ by directly imposing a maximum-entropy regularization on the output of the pathway generator:

$$\mathcal{L}_{\text{explore}} = -\mathcal{H}\left(\mathbb{E}_{(\mathbf{x},\mathbf{y})\sim\mathcal{D}}G(\mathbf{x})\right), \tag{4}$$

where $\mathcal{H}(\mathbf{z}) = -\sum_i z_i \cdot \log(z_i)$ is the entropy function. The loss $\mathcal{L}_{\text{explore}}$ maximizes the entropy of the average pathway weights of all the target data. It enforces that from the viewpoint of the whole dataset, all models have the chance to be activated and different pathways over the pre-trained models are fully explored if they truly promote the target performance, thereby avoiding the hub collapse.

## 3.2 Enhancing Exploitation in Activated Models

With the above pathway designs and losses, we can learn better data-dependent pathways for each piece of target data. With the derived pathways, we know which pre-trained models to transfer knowledge from, and there remains the question of how to fully exploit such knowledge. The task-specific loss $\mathcal{L}_{\text{task}}$ utilizes and updates the knowledge in activated models, but each model contributes only a fraction to such prediction and optimization process. To further enhance the transfer of knowledge and ensure the performance of the activated models, we further tune the pre-trained models $\Theta_i|_{i=1}^m$ with the task-specific loss on the specific data that activate them:

$$\mathcal{L}_{\text{exploit}} = \sum_{i=1}^m \mathbb{E}_{(\mathbf{x},\mathbf{y}) \sim \mathcal{D}} \mathbb{I}\left(\bar{G}(\mathbf{x})_i > 0\right) \ell\left(\Theta_i(\mathbf{x}), \mathbf{y}\right). \tag{5}$$

The above mechanism of further exploiting more useful models in the hub for each datum shares similar spirit with reinforcement learning in exploiting the more rewarded states in the environments.

Empowering Hub-Pathway with exploration and exploitation, the final optimization problem becomes

$$\begin{aligned} \arg\min_{G,A} \mathcal{L}_{\text{task}} + \lambda \cdot \mathcal{L}_{\text{explore}} \\ \arg\min_{\Theta_i|_{i=1}^m} \mathcal{L}_{\text{task}} + \mathcal{L}_{\text{exploit}}, \end{aligned} \tag{6}$$

where we use a trade-off $\lambda$ between the target-task-specific loss and the exploration-encouraging loss since they have different scales. $\mathcal{L}_{\text{task}}$ and $\mathcal{L}_{\text{exploit}}$ are both derived from the target-task-specific loss with the same scale, and thus no trade-off is needed. Our Hub-Pathway framework simultaneously explores data-dependent pathway configurations throughout the model hub and exploits the activated models by aggregating their useful knowledge and enhancing knowledge transfer to downstream tasks. The framework can be trained in an end-to-end way. The computation cost at both the training and testing time is at most $k$ times of the single model fine-tuning, which is much more efficient than a naive ensemble of multiple models with the computation cost linearly increasing with the hub size.

## 4 Experiments

To verify that Hub-Pathway framework can address the general model hub transfer learning problem, we conduct experiments on tasks including image classification, facial landmark detection and reinforcement learning, which are proposed in [59]. In addition to the model hub with the homogeneous architecture evaluated by prior works, we also conduct experiments on different downstream tasks with the model hub consisting of pre-trained models with heterogeneous architectures, which is a more general situation that cannot be solved effectively by existing works.

### 4.1 Transfer Learning in Image Classification

**Homogeneous Model Hub.** Previous work of Zoo-Tuning [59] explores a hub of models with the *homogeneous* architecture. We follow this setting and use 5 ResNet-50 models pre-trained on 5 widely-used computer vision datasets: (1) Supervised pre-trained model and (2) Unsupervised pre-trained model with MOCO [26] on ImageNet [55], (3) Mask R-CNN [28] model for detection and instance segmentation, (4) DeepLabV3 [8] model for semantic segmentation, and (5) Keypoint R-CNN model for keypoint detection, pre-trained on COCO-2017 challenge datasets of each task. All pre-trained models are drawn from torchvision model hub [50] or from the original implementation.

**Heterogeneous Model Hub.** To verify that Hub-Pathway could address more general model hub transfer learning settings, we further conduct experiments on a model hub consisting of models with *heterogeneous* architectures. We consider a model hub consisting of some other famous architectures including MobileNetV3 [32], EfficientNet [61], Swin Transformer [43] and ConvNeXt [44]. For the MobileNetV3 architecture, we use the MobileNetV3-Large-1.0 pre-trained model, which achieves the best performance in this model family. For the last three architectures, we use EfficientNet-B3 (EffNet-B3), Swin-Transformer-Tiny (Swin-T) and ConvNeXt-Tiny (ConvNeXt-T), which achieve comparable performance when pre-trained on ImageNet. We also utilize the Mask R-CNN model mentioned above to further increase the diversity of the training data and tasks in the hub.

**Benchmarks.** We verify the efficacy of Hub-Pathway on 7 various classification tasks evaluated in [59], including: **(1)** *General* classification benchmarks with **CIFAR-100** [39] and **COCO-70** [70]; **(2)** *Fine-grained* benchmarks for aircraft (**FGVC Aircraft** [45]), car (**Stanford Cars** [38]) and indoor

Table 1: Results on the classification benchmarks with the homogeneous model hub.

| Model | General | | Fine-Grained | | | Specialized | | Avg. |
|---|---|---|---|---|---|---|---|---|
| | CIFAR | COCO | Aircraft | Cars | Indoors | DMLab | EuroSAT | |
| ImageNet | 81.18 | 81.97 | 84.63 | 89.38 | 73.69 | 74.57 | 98.43 | 83.41 |
| MoCo | 75.31 | 75.66 | 83.44 | 85.38 | 70.98 | 75.06 | 98.82 | 80.66 |
| MaskRCNN | 79.12 | 81.64 | 84.76 | 87.12 | 73.01 | 74.73 | 98.65 | 82.72 |
| DeepLab | 78.76 | 80.70 | 84.97 | 88.03 | 73.09 | 74.34 | 98.54 | 82.63 |
| Keypoint | 76.38 | 76.53 | 84.43 | 86.52 | 71.35 | 74.58 | 98.34 | 81.16 |
| Ensemble | 82.26 | 82.81 | 87.02 | 91.06 | 73.46 | 76.01 | 98.88 | 84.50 |
| Distill | 82.32 | 82.44 | 85.00 | 89.47 | 73.97 | 74.57 | 98.95 | 83.82 |
| K-Flow | 81.56 | 81.91 | 85.27 | 89.22 | 73.37 | 75.55 | 97.99 | 83.55 |
| ModelSoups | 81.32 | 82.94 | 85.24 | 90.32 | 75.61 | 74.29 | 98.65 | 84.05 |
| Zoo-Tuning-L | 83.39 | 83.50 | 85.51 | 89.73 | 75.12 | 75.22 | **99.12** | 84.51 |
| Zoo-Tuning | **83.77** | **84.91** | 86.54 | 90.76 | 75.39 | 75.64 | **99.12** | 85.16 |
| Hub-Pathway | 83.31 | 84.36 | **87.52** | **91.72** | **76.91** | **76.47** | **99.12** | **85.63** |

scene (**MIT-Indoors** [51]) classification; **(3)** *Specialized* benchmarks collected from the DeepMind Lab environment (**DMLab** [4]) and Sentinel-2 satellite (**EuroSAT** [30]). We follow the common fine-tuning principle and replace the last layer with a randomly initialized fully connected layer [69]. We conduct each experiment 3 times with different seeds and report the average top-1 accuracy.

**Implementation Details.** We follow the standard fine-tuning principle in [69]. The source task-specific heads in the pre-trained models are replaced with newly initialized fully connected layers as the target task-specific heads. We adopt the SGD optimizer with an initial learning rate of $0.01$ and momentum of $0.9$. The models are trained for 15k iterations with a batch size of $48$. The learning rate is decayed by a rate of $0.1$ at the 6k-th and 12k-th iterations. We fine-tune the pre-trained models and train the pathway generator and aggregator simultaneously on the target data. The pathway generator is implemented as a 9-layer ResNet [29]. We run the experiments 3 times and report the mean results of top-1 accuracy. For all the datasets, we follow the same dataset split as in [59].

**Results on Homogeneous Model Hub.** We show the results on all the benchmarks in Table 1. We compare with fine-tuning from each single pre-trained model, the ensemble by voting of the predictions of different pre-trained models after fine-tuning (Ensemble), knowledge distillation from the Ensemble model (Distill) and model zoo transfer learning methods including: Knowledge Flow [41] (K-Flow), ModelSoups [64], Zoo-Tuning-L and Zoo-Tuning [59]. For our method, we activate top-2 models for each datum in all our experiments. Hub-Pathway outperforms all the compared methods. In particular, different pre-trained models also show different transfer performances on different benchmarks, which demonstrates that different pre-trained models have different transferability to different target tasks. We can also get the results of a 'Domain Expert' baseline which chooses the best single model for each task. We observe that methods using all the pre-trained models outperform Domain Expert, which demonstrates that the model hub contains much more abundant knowledge than a single model and proves the practical use of the model hub transfer learning setting. Hub-Pathway and Zoo-Tuning both outperform Ensemble and Distill, which demonstrates that a simple ensemble of pre-trained models may not fully utilize the knowledge in the model hub and also may cause different pre-trained models to influence each other. K-Flow and Zoo-Tuning-L perform similarly to Ensemble, which shows that a task-level model aggregation strategy does not maximally utilize the knowledge in the model hub. Finally, our Hub-Pathway outperforms Zoo-Tuning and Zoo-Tuning-L even though Zoo-Tuning is specially designed to address the situation that all the pre-trained models have the same architecture. The observation demonstrates that Hub-Pathway finds more transferable models and better transfer knowledge from them.

**Results on Heterogeneous Model Hub.** Previous work of Knowledge Flow requires manually designing the connections of different layers across different models and applying feature transformation between them. It cannot work in such a model hub with heterogeneous architectures since the correspondence between layers of different pre-trained models is unknown, and their features have different shapes and structures. Zoo-Tuning has an even stricter constraint of the same architecture of the models in the hub, which limits its application in such a more general situation. Therefore, we only compare our method with single-model transfer learning, Ensemble and Distill. From Table 2,

Table 2: Results on the classification benchmarks with the heterogeneous model hub.

| Model | General | | Fine-Grained | | | Specialized | | Avg. |
|---|---|---|---|---|---|---|---|---|
| | CIFAR | COCO | Aircraft | Cars | Indoors | DMLab | EuroSAT | |
| MaskRCNN | $79.12_{\pm0.06}$ | $81.64_{\pm0.39}$ | $84.76_{\pm0.30}$ | $87.12_{\pm0.09}$ | $73.01_{\pm0.45}$ | $74.73_{\pm0.46}$ | $98.65_{\pm0.05}$ | 82.72 |
| MobileNetV3 | $83.14_{\pm0.10}$ | $83.28_{\pm0.05}$ | $80.26_{\pm0.03}$ | $86.37_{\pm0.61}$ | $75.09_{\pm0.19}$ | $70.09_{\pm0.24}$ | $98.95_{\pm0.11}$ | 82.45 |
| EffNet-B3 | $87.28_{\pm0.21}$ | $86.97_{\pm0.08}$ | $83.99_{\pm0.09}$ | $89.34_{\pm0.13}$ | $78.16_{\pm0.16}$ | $72.69_{\pm0.27}$ | $99.13_{\pm0.01}$ | 85.37 |
| Swin-T | $84.37_{\pm0.12}$ | $84.12_{\pm0.01}$ | $80.82_{\pm0.27}$ | $89.10_{\pm0.09}$ | $73.39_{\pm0.34}$ | $72.22_{\pm0.24}$ | $98.69_{\pm0.05}$ | 83.24 |
| ConvNeXt-T | $86.96_{\pm0.10}$ | $87.15_{\pm0.09}$ | $84.23_{\pm0.57}$ | $90.67_{\pm0.04}$ | $81.66_{\pm0.07}$ | $73.80_{\pm0.11}$ | $98.65_{\pm0.04}$ | 86.16 |
| Ensemble | $87.72_{\pm0.19}$ | $88.04_{\pm0.07}$ | $87.11_{\pm0.28}$ | $92.68_{\pm0.33}$ | $82.79_{\pm0.26}$ | $\mathbf{74.86}_{\pm0.14}$ | $99.23_{\pm0.01}$ | 87.49 |
| Distill | $87.33_{\pm0.16}$ | $88.09_{\pm0.25}$ | $85.26_{\pm0.32}$ | $91.39_{\pm0.19}$ | $81.51_{\pm0.29}$ | $74.75_{\pm0.20}$ | $99.24_{\pm0.02}$ | 86.80 |
| Hub-Pathway | $\mathbf{89.01}_{\pm0.06}$ | $\mathbf{89.14}_{\pm0.12}$ | $\mathbf{88.12}_{\pm0.14}$ | $\mathbf{92.93}_{\pm0.20}$ | $\mathbf{84.40}_{\pm0.22}$ | $74.80_{\pm0.23}$ | $\mathbf{99.26}_{\pm0.06}$ | $\mathbf{88.24}$ |

we observe that Hub-Pathway outperforms all these methods in most of the tasks. Note that, with the activation of 2 models for each input, the *training and inference cost* of Hub-Pathway is much less than the strong Ensemble method. The observation demonstrates that compared with existing methods, Hub-Pathway is an effective and efficient solution to more general situations.

## 4.2 Transfer Learning in Facial Landmark Detection

We conduct experiments on the Facial Landmark Detection datasets to further demonstrate that Hub-Pathway can be applied to a wide range of applications. We use the same model hub as the image classification setting in Section 4.1 and transfer to three facial landmark detection tasks: **300W** [57], **WFLW** [66], and **COFW** [7], which creates a large domain gap between pre-trained models and the target task. We generally follow the protocol in [60] and the standard training scheme in [66]. In testing, each keypoint location is predicted by transforming the highest heat value location to the original image space and adjusting it with a quarter offset in the direction from the highest response to the second highest response [12].

**Implementation Details.** For the facial landmark detection experiments, we generally follow the training and testing protocols in [60] and the standard training scheme in [66]. The models are trained for 60 epochs with a batch size of 16 using the Adam optimizer. The learning rate is set as 0.0001 initially and is decayed by a rate of 0.1 at the 30-th and 50-th epochs. In testing, each keypoint location is predicted by transforming the highest heat value location to the original image space and adjusting it with a quarter offset in the direction from the highest response to the second highest response [12]. We follow the same data split and pre-processing as in [60]. All the faces are cropped by the provided boxes according to the center location and resized to $256 \times 256$. We augment the data by $\pm30$ degrees in-plane rotation, $0.75 \sim 1.25$ scaling, and random flipping.

**Results.** We use the inter-ocular distance as normalization and report the normalized mean error (NME) for evaluation in Table 3, where smaller values mean better performance. We observe that fine-tuning from a single pre-trained model outperforms training from scratch, which demonstrates that knowledge transfer generally improves the target performance even though the pre-trained models are trained from different tasks. K-Flow and Zoo-Tuning perform worse than Ensemble with a non-trivial margin in most of the tasks, which demonstrates that they fail to extract and transfer generalizable knowledge and thus do not perform well under a large domain gap between the pre-trained models and the target task. Hub-Pathway instead outperforms Ensemble on the WFLW task and achieves a comparable performance to Ensemble on the 300W and COFW. It is worth noting that Hub-Pathway is much more efficient in computation and memory than Ensemble. The results demonstrate that Hub-Pathway could generalize across tasks even with large domain gaps.

Table 3: NME results on facial landmark detection tasks. Smaller values mean better performance.

| Model | 300W | WFLW | COFW |
|---|---|---|---|
| From Scratch | 3.66 | 5.33 | 4.20 |
| ImageNet | 3.52 | 4.90 | 3.66 |
| MoCo | 3.45 | 4.75 | 3.63 |
| MaskRCNN | 3.53 | 4.87 | 3.67 |
| DeepLab | 3.53 | 4.89 | 3.73 |
| Keypoint | 3.50 | 4.90 | 3.66 |
| Ensemble | **3.33** | 4.64 | **3.46** |
| Distill | 3.45 | 4.74 | 3.53 |
| K-Flow | 3.71 | 5.28 | 4.58 |
| Zoo-Tuning | 3.41 | 4.58 | 3.51 |
| Hub-Pathway | **3.33** | **4.54** | 3.51 |

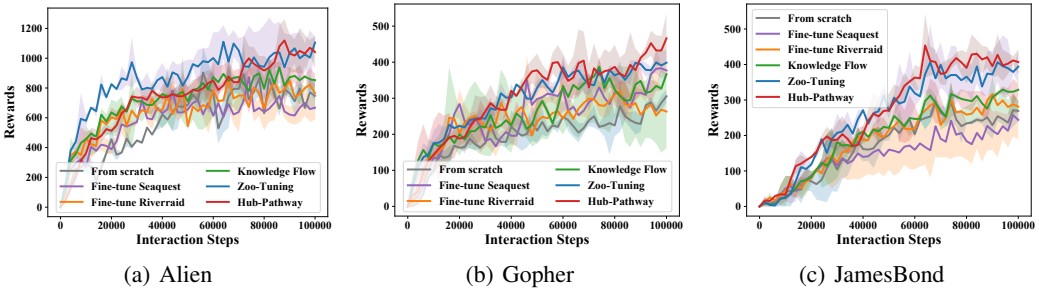

Figure 2: Results of model hub transfer learning to three reinforcement learning tasks in Atari games.

## 4.3 Transfer Learning in Reinforcement Learning

To demonstrate the generalizability of the proposed Hub-Pathway framework to various domains and tasks, we further conduct experiments on reinforcement learning models.

**Benchmarks.** Current reinforcement learning algorithms have already achieved human-level performance on Atari games, but suffer from the low data efficiency with the requirement of a large amount of interaction steps. So we measure our method at 100k interaction steps on Atari, which is similar to the time to train a human learner. We use the Seaquest and Riverraid tasks as source tasks and construct the model hub consisting of optimal policies for the source tasks. We then transfer to 3 downstream tasks: Alien, Gopher, and JamesBond.

**Implementation Details.** For the learning algorithm, we follow the implementation of Data-Efficient Rainbow [63], which modifies hyper-parameters of Rainbow [31] for data efficiency. The model of Data-Efficient Rainbow consists of 2 convolutional layers: 32 filters of size $5 \times 5$ with the stride of 5 and 64 filters of size $5 \times 5$ with the stride of 5, followed by a flatten layer and 2 noisy linear layers [22] with the hidden size of 256. We use Adam optimizer [36] with a learning rate of $1 \times 10^{-4}$. Other hyper-parameters are kept the same as those in [63]. We repeat each experiment 5 times with different seeds and report the mean and variance of the results.

**Results.** We show the transfer learning performance to the three downstream tasks in Figure 2. We compare with several methods, including training from scratch, fine-tuning every single model, and two multi-model transfer learning methods: Knowledge Flow [41] and Zoo-Tuning [59]. Compared with Zoo-Tuning, which is a method tailored for this setup, Hub-Pathway slightly outperforms it on Gopher and achieves comparable performance on Alien and JamesBond. Note that our method is generally applied to various architectures of pre-trained models but still works well in the homogeneous architecture setting. In particular, fine-tuning from a single model converges to the same or even worse reward than training from scratch, which demonstrates that brutely fine-tuning does not improve the performance in reinforcement learning. Our Hub-Pathway framework fully utilizes the transferred knowledge from the pre-trained policies and thus improves the target task performance.

## 4.4 Analysis

**Analysis on Pathway Weights.** We visualize the average pathway weight of each pre-trained model on all the target data in different datasets in Figure 3(a). We observe that for different datasets, the distribution of average pathway weight across all the pre-trained models are different. The observation demonstrates the challenge that different pre-trained models have different transferability to the target task, and the proposed Hub-Pathway could assign different pathway weights to different models.

The above results demonstrate that Hub-Pathway assigns diverse weights to different pre-trained models. To further verify the quality of weights, we fine-tune all the pre-trained models on the target dataset and select the samples that can be classified correctly by at least one of the models in the model hub from each dataset. Then an accurate pathway weight should assign the highest pathway weight to at least one of the models with the correct prediction. For each sample, we make the prediction in two ways: one is to use the model with the highest pathway weight and the other is to use a randomly selected model from the model hub. We report the prediction accuracy of these two ways on the selected samples in different datasets in Figure 3(b). The results show that the selected

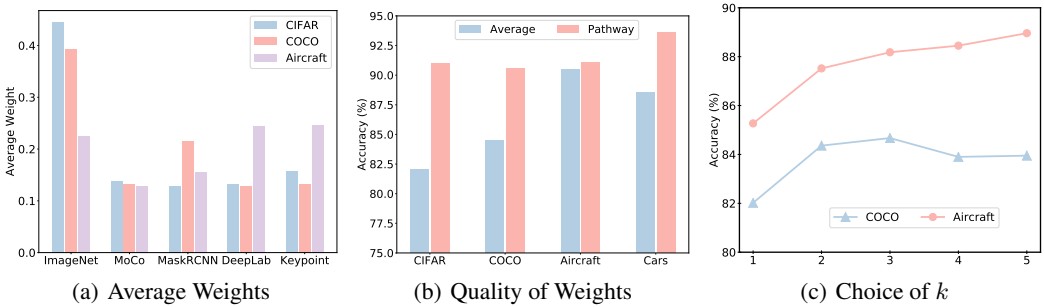

|           (a) Average Weights           |           (b) Quality of Weights           |           (c) Choice of $k$           |

Figure 3: (a) The average pathway weights of different models for different datasets; (b) The quality of the learned pathway weights; (c) The target performance with respect to different choices of $k$.

model with the highest pathway weight performs much better than a uniformly sampled model, which indicates that the learned pathway can find which pre-trained models are more transferable.

**Visualization of Pathway Weights.** We visualize some images from the COCO-70, Aircraft, Stanford Cars and MIT-Indoors datasets along with their pathway weights assigned by the trained Hub-Pathway in Figure 4. We observe that the proposed Hub-Pathway can truly learn some data-dependent pathway weights. Overall, different data can be assigned with diverse pathway weights to use the most transferable knowledge in the model hub. Each model has the chance to play their roles in prediction, which enables fully using the rich knowledge in the hub. Besides, some similar samples (such as the left two images on the first row) may have similar pathway weights. We also observe that the ImageNet pre-trained model generally contributes more to these samples from object recognition tasks, which is also consistent with our understanding of the relationship between pre-training and downstream tasks.

**Variants of Hub-Pathway.** We conduct experiments on the variants of Hub-Pathway to verify the efficacy of each component in our framework: (1) Hub-Pathway w/ random path replaces the learned data-dependent pathway with a random pathway, which aims to verify the learned pathway actually finds the more transferable models; (2) Hub-Pathway w/o explore removes the loss $\mathcal{L}_{\text{explore}}$, which may learn pathways discarding some pre-trained models; (3) Hub-Pathway w/o exploit removes the loss $\mathcal{L}_{\text{exploit}}$, which does not fine-tune the activated models with the exploit loss. We conduct experiments by transferring to the tasks of COCO and Cars. The results are shown in Table 4. We observe that Hub-Pathway outperforms Hub-Pathway w/o explore,

demonstrating that it is important to explore more pathway configurations. Hub-Pathway outperforms Hub-Pathway w/o exploit, showing that fine-tuning the activated models encourages knowledge transfer to the target task. Hub-Pathway outperforms Hub-Pathway w/ random path, which demonstrates that the proposed framework learns a non-trivial pathway to select the most transferable pre-trained models for the target data prediction.

Table 4: Ablation study on Hub-Pathway.

| Method | COCO | Cars |
|---|---|---|
| Pathway w/ random path | 80.97 | 90.61 |
| Pathway w/o explore | 81.34 | 89.75 |
| Pathway w/o exploit | 80.10 | 90.59 |
| Hub-Pathway | 84.36 | 91.72 |

**Choice of $k$ in $f_{\text{topk}}$.** Our framework only activates the models within the highest top $k$ pathway weights. We investigate the influence of the value $k$ on the final target performance in Fig 3(c). We observe that the performance does not change rapidly with respect to $k$ except for $k = 1$, which means that aggregating the knowledge from different pre-trained models is important. It indicates that we can use the Hub-Pathway framework for efficient transfer learning from the model hub.

**The Need of Data-Dependent Pathways.** Switching models in the dataset or task level cannot fully use the knowledge in the model hub since different target data may have different relations to pre-trained models even in the same dataset. As shown in Table 5, we report the prediction performance of the best model for each dataset (Best Single), the ensemble of the best $k$ models for each task (Ensemble Top-k, $k$=2), and using the best model for each data point (Oracle). Note that here both Best Single and Ensemble Top-k choose models at the task level, and we use the same

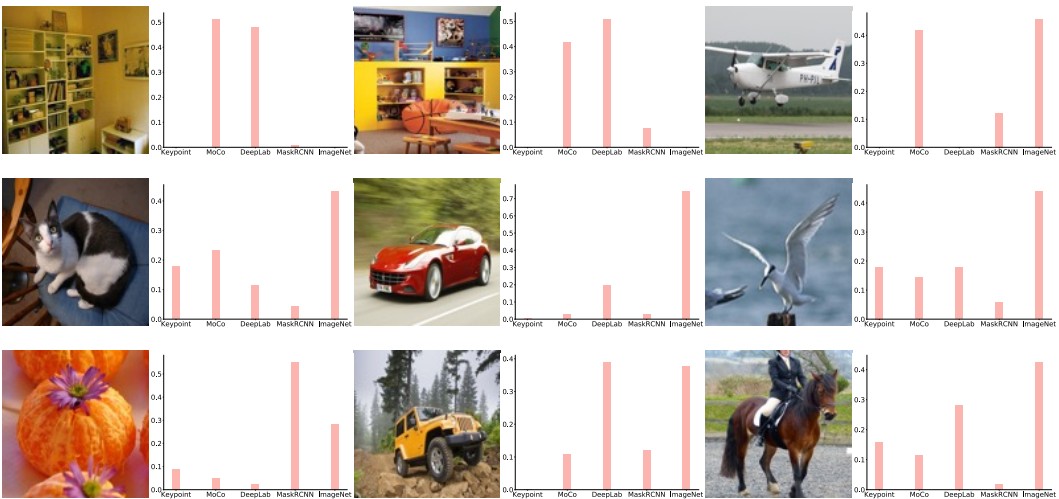

Figure 4: Visualization of test samples along with their pathway weights.

Table 5: Ablation results on the classification benchmarks with the homogeneous model hub.

| Model | General | | Fine-Grained | | | Specialized | | Avg. |
| | CIFAR | COCO | Aircraft | Cars | Indoors | DMLab | EuroSAT | |
|---|---|---|---|---|---|---|---|---|
| Best Single | 81.18 | 81.97 | 84.97 | 89.38 | 73.69 | 75.06 | 98.82 | 83.58 |
| Ensemble Top-k | 82.85 | 83.24 | 85.55 | 90.05 | 74.16 | 75.48 | 98.83 | 84.31 |
| Hub-Pathway | 83.31 | 84.36 | 87.52 | 91.72 | 76.91 | 76.47 | 99.12 | 85.63 |
| Oracle | 88.52 | 89.17 | 92.84 | 94.86 | 83.91 | 84.71 | 99.37 | 90.48 |

model(s) to predict all the test data in the task. While for Oracle, we predict each test data point with the oracle model performing the best on this data point (the oracle model is chosen by comparing the model outputs with the true label of each test data point), which is a more fine-grained instance-level selection. We observe that Oracle outperforms both Ensemble Top-k and Best Single with a large margin, which indicates that the instance-level model selection is important and the data-dependent pathway is necessary. Also, we observe that Hub-Pathway outperforms Ensemble Top-k, which shows that Hub-Pathway is a good attempt to explore data-dependent pathways.

# 5 Conclusion

In this paper, we propose a Hub-Pathway framework to enable knowledge transfer from a hub of diverse pre-trained models. We design an architecture with data-dependent pathways to control which pre-trained models are used for the current data at the input level and aggregate the outputs from activated pre-trained models at the output level. The framework is optimized by a task-specific loss to ensure that the pathways introduce high transfer performance. We employ a noisy pathway generator and design an exploration loss to encourage the framework to explore and compare different pathways. To promote the exploitation of the knowledge in the activated models, we further fine-tune the models with the specific data activating them. Experimental results on model hubs with homogeneous and heterogeneous architectures demonstrate that the proposed framework achieves the state-of-the-art performance for model hub transfer learning.

# Acknowledgments

This work was supported by the National Key Research and Development Plan (2021YFB1715200), National Natural Science Foundation of China (62022050 and 62021002), Beijing Nova Program (Z201100006820041), BNRist Innovation Fund (BNR2021RC01002), and Huawei Innovation Fund.

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
