# OpenReview forum: "Hub-Pathway: Transfer Learning from A Hub of Pre-trained Models"
_NeurIPS.cc/2022/Conference — NeurIPS 2022 Accept_

### Official Review · Reviewer_AErg · 2022-07-06

**Rating:** 7
**Confidence:** 5
**Soundness:** 4 excellent
**Presentation:** 3 good
**Contribution:** 3 good

**Summary:**

This study proposes Hub-Pathway, a method for maximizing the knowledge gained from multiple pre-trained deep neural network models in a transfer learning problem setting. The basic idea of the Hub-pathway is to select the best subset of models for each datum from the model hub to make predictions. There are two main challenges of this approach: (i) how to select the best path for each data set and (ii) how to aggregate knowledge from multiple models. For the first challenge, the paper introduces a gating function with randomness to achieve a variety of path options for each datum. For the second challenge, the paper develops a mechanism that combines the aforementioned gate function with a function for knowledge aggregation to output a prediction result for the hub as a whole. In the experiments, the paper compared the proposed method with prior methods in the tasks of classification, facial landmark detection, and reinforcement learning, and showed that the proposed method achieves superior performances by dynamically routing paths on model hubs.

**Questions:**

### Q1. Do we need to change the path for each datum?
One of the key ideas in this paper is to efficiently extract knowledge from pre-trained models by switching paths for each datum. We have a concern on this point. That is, wouldn't it be sufficient to just select a model for each dataset? We could not validate the proposed method of switching on each datum because the evaluation in the current paper analyzed only the performance for each target dataset. Rather than holding a large number of models and training them simultaneously, a more realistic setting in terms of inference cost would be to test a set of fine-tuned models on the target dataset and select Top-K models by validation scores to ensemble. In order to argue for the validity of switching on each datum, we believe that a comparison with such a method of switching models on each target dataset is necessary. This evaluation can easily be done using the single model that appeared in Tables 1 and 2.

### Q2. How is the performance of the proposed method compared to ModelSoups on homogeneous?
This is related to Q1. In the homogeneous setting, a recent paper presented a method to aggregate the knowledge in multiple models called ModelSoups[1], which averages weights of fine-tuned models and applies the averaged weights to a single model. ModelSoups also adopts a Top-K selection technique as well as hub-pathway and thus it can be considered a model aggregating knowledge for each target dataset level discussed above. Obviously, we recognize that ModelSoups is not a perfect competitor to hub-pathway since it cannot be applied to heterogeneous settings, but we consider a comparison with ModelSoups would be helpful for investigating the characteristics of hub-pathway.

```
[1] Wortsman, Mitchell, et al. "Model soups: averaging weights of multiple fine-tuned models improves accuracy without increasing inference time." International Conference on Machine Learning (2022).
```

### Q3. About experimental settings of the image classification task
In the image classification experiments, the number of training epochs for hub-pathway and the other competitors did not seem to be described. Does hub-pathway require more training time than other methods?


**Limitations:**

Nothing to report.

**Strengths And Weaknesses:**

### Strength
* Research questions and motivations are clear.
* The proposed method is quite simple so it is easy to reimplement.
* The proposed method can handle heterogeneous sets of models, unlike prior methods.
* Comparison with prior methods in a variety of experimental settings.

### Weakness
* Insufficient explanation and evaluation of the need to change paths for each datum.
* A little lack of evaluation in a homogenous setting.
* Larger memory size and execution time required during inference time.

The paper is well-structured and clearly describes the research questions and motivations. The proposed method provides new insights into this research area by achieving knowledge aggregation from heterogeneous model hubs, which has been difficult before, and by showing better experimental results than models in homogeneous settings. On the other hand, comparisons with state-of-the-art methods and verification of the validity of switching paths on a datum-by-datum basis are insufficient (these will be discussed in Questions). In addition, the proposed method arises new challenges in terms of memory size and execution time during inference. However, these issues might not be considered serious enough flaws to damage the main argument of the paper.

---

> ### Author Response · Authors · 2022-08-02
> **Response to Reviewer AErg**
>
> Many Thanks to Reviewer AErg for providing a detailed review and insightful questions.
>
> **Q1:** The need of the data-dependent pathway.
>
> Switching models in the dataset or task level cannot fully use the knowledge in the model hub since different target data may have different relations to pre-trained models even in the same dataset. As shown in the table below, we report the prediction performance of the best model for each dataset (Best Single), the ensemble of the best k models for the task (Ensemble Top-k, k=2), and using the best model for each data point (Oracle). Note that here both Best Single and Ensemble Top-k choose models at task-level and we use the same model(s) to predict all the data in the task. While for Oracle, we predict each data point with the oracle model performing the best on this data point (the oracle model is chosen by comparing the model outputs with the true label of each data point), which is a more fine-grained instance-level selection. We observe that Oracle outperforms both Ensemble Top-k and Best Single with a large margin, which indicates that the instance-level model selection is important and the data-dependent pathway is necessary. Also, we observe that Hub-Pathway outperforms Ensemble Top-k, which shows that Hub-Pathway is a good attempt to explore data-dependent pathways.
>
> | Method | CIFAR | COCO | Aircraft | Cars | Indoors | DMLab | EuroSAT | Avg. |
> | ------ | ------ | ------ | ------ | ------ | ------ | ------ | ------ | ------ |
> | Best Single | 81.18 | 81.97 | 84.97 | 89.38 | 73.69 | 75.06 | 98.82 | 83.58 |
> | Ensemble Top-k | 82.85 | 83.24 | 85.55 | 90.05 | 74.16 | 75.48 | 98.83 | 84.31 |
> | Hub-Pathway | 83.31 | 84.36 | 87.52 | 91.72 | 76.91 | 76.47 | 99.12 | 85.63 |
> | Oracle | 88.52 | 89.17 | 92.84 | 94.86 | 83.91 | 84.71 | 99.37 | 90.48 |
>
> **Q2:** The performance of ModelSoups.
>
> ModelSoup averages the weights of different fine-tuned models. We conduct experiments on two different ways of creating the model soup. The first way is ModelSoups-Same, where we fine-tune from ImageNet pre-trained model multiple times with different hyper-parameter settings and select the top 2 models to average as the model soup. The second way is ModelSoups-Different, where we fine-tune from five different pre-trained models shown in Table 1 and select the top 2 models to average as the model soup. We report the results in the table below. We observe that Hub-Pathway outperforms both model soup configurations. ModelSoups-Different fails, and we think it could be because models fine-tuned from different pre-trained models may fall into diverse local minimals, which is not suitable for parameter averaging. We add the ModelSoups-Same results in $\underline{\text{Table 1 of the revised paper}}$.
>
> | Method | CIFAR | COCO | Aircraft | Cars | Indoors | DMLab | EuroSAT | Avg. |
> | ------ | ------ | ------ | ------ | ------ | ------ | ------ | ------ | ------ |
> | ModelSoups-Same | 81.32 | 82.94 | 85.24 | 90.32 | 75.61 | 74.29 | 98.65 | 84.05 |
> | ModelSoups-Different | 70.27 | 79.43 | 64.90 | 68.51 | 66.42 | 64.72 | 94.67 | 72.70 |
> | Hub-Pathway | 83.31 | 84.36 | 87.52 | 91.72 | 76.91 | 76.47 | 99.12 | 85.63 |
>
> **Q3:** The experimental settings of the image classification task.
>
> We plot the training curve for the Cars task in the image classification dataset in $\underline{\text{Figure 2(b) of the revised appendix}}$. We compare the convergence speed of Hub-Pathway and a single ImageNet model. We observe that in the first stage, Hub-Pathway converges a little slower because of pathway finding, but then it converges with a similar speed to the single ImageNet model. As stated in the implementation details, Hub-Pathway and competitors are trained for the same iterations.

---

> > ### Comment · Reviewer_AErg · 2022-08-05
> > **Thank you for your responses**
> >
> > ## Reply to the response for Q1
> > Thank you for the detailed response supported by the clear results. I think the explanation and experiment further strengthen the justification for selecting paths data-dependently. This concern is addressed by your response. Please add this experiment to the paper (or appendix) as well.
> >
> > ## Reply to the response for Q2
> > We thank you for presenting the results of the comparative experiments on ModelSoup despite the short period. We have a question about ModelSoup-Different, how do you average weights with different architectures? Our understanding is that ModelSoup can only be naively applied in a homogeneous setting (i.e., ModelSoups-Same) with a shared architecture.
> >
> > ## Reply to the response for Q3
> > Thank you for the additional results and explanations. This concern is addressed by your response.

---

> > > ### Author Response · Authors · 2022-08-05
> > > **Response to Reviewer AErg**
> > >
> > > Thank you for the comments.
> > >
> > > **Q1:** The need of the data-dependent pathway.
> > >
> > > We have added this experiment to $\underline{\text{Table 5 of the revised appendix}}$ to strengthen the motivation for selecting data-dependent pathways.
> > >
> > >
> > > **Q2:** Clarification on ModelSoups-Same and ModelSoups-Different.
> > >
> > > Both ModelSoups-Same and ModelSoups-Different are performed on models with the same architecture. In ModelSoups-Different, 'different' means that the models are pre-trained from different datasets or tasks.
> > >
> > > To be specific, in ModelSoups-Same, we fine-tune from the same ImageNet pre-trained model (with the architecture ResNet-50) multiple times with different hyper-parameters to derive multiple fine-tuned models. Then we average the parameters of the top 2 models among them to derive the results of ModelSoups-Same. This is the protocol of how ModelSoups is used in their original paper.
> > >
> > > Since in our paper, we consider transfer learning from a hub of models pre-trained from different sources, so we extend the idea of parameter averaging and explore a variant of ModelSoups-Different. In ModelSoups-Different, we fine-tune from the five different pre-trained models: ImageNet, MoCo, MaskRCNN, DeepLab, Keypoint in $\underline{\text{Table 1}}$ (all with the architecture ResNet-50), to derive multiple fine-tuned models. Then we average the parameters of the top 2 models among them to derive the results of ModelSoups-Different.
> > >
> > > From the reported results, we can find that ModelSoups is a good fine-tuning strategy to promote performance over vanilla fine-tuning. However, it may not be directly used for transfer learning with models from different sources since ModelSoups-Different does not achieve good performance. We think the reason is that although with the same architecture, models fine-tuned from different pre-trained models may fall into diverse local minimals, which is unsuitable for direct parameter averaging.

---

> > > > ### Comment · Reviewer_AErg · 2022-08-05
> > > > **Thank you for your responses again**
> > > >
> > > > **Q1**: The need of the data-dependent pathway.
> > > >
> > > > > We have added this experiment to Table 5 of the revised appendix
> > > >
> > > > We found that. Thank you.
> > > >
> > > > **Q2**: Clarification on ModelSoups-Same and ModelSoups-Different.
> > > >
> > > > Thank you for the detailed explanations. Your explanation of the ModelSoups-Different setup makes sense to us. Since the parameter averaging of ModelSoup implicitly assumes that the models have the loss surfaces for each other, so it is convincing that it wouldn't work when averaging models pre-trained on different datasets.
> > > >
> > > > Finally, we would raise our score because all of our concerns were addressed by the author's responses and we have increased confidence that this paper should be presented at the conference.

---

> > > > > ### Author Response · Authors · 2022-08-05
> > > > > **Many thanks for your comments**
> > > > >
> > > > > We would like to sincerely thank Reviewer AErg for acknowledging our contributions and providing detailed and insightful reviews, which inspire us to strengthen our motivation for data-dependent pathways and explore the related work of ModelSoups. Also, many thanks for the careful judgment of our feedback and the timely responses.

---

### Official Review · Reviewer_8MEA · 2022-07-07

**Rating:** 6
**Confidence:** 3
**Soundness:** 3 good
**Presentation:** 3 good
**Contribution:** 2 fair

**Summary:**

This paper tackles the problem of transfer learning from a zoo / hub of pre-trained models unto a specific end-task.
They treat the problem as data-point dependent and route each data-point through an adaptive subset of the available pre-trained models.
They also fine-tune the pre-trained models on the data-sets that activate them.
This paper compares to some relevant baselines and shows improvement over them.

**Questions:**

1. Could you provide error bars on your experiments since you have 3 runs per experiment
2. Are all the inputs similarly structured ? Do all the models expect the same type of input structure - no discussion of adapting model input to task structure.
3. Why add to the computational cost by using a randomized sub-network G_n -> why not just do something like temperature annealing ?
4. How about a domain expert baseline ? Assuming oracle access to the best single model apriori - can you update the tables with what the performance would be ? This is the baseline of an expert who knows how to pick the best single model for each new task.
5. For the RL experiments (Fig 2), the author say "We observe that the proposed Hub-Pathway outperforms these state-of-the-art methods" - however the errorbars with the ZooTuning experiments clearly overlap. Am I missing something ?

**Limitations:**

1. The method introduces extra hyper-parameters over methods like ensembling

**Strengths And Weaknesses:**

**Strengths**
1. Method works for heterogeneous architectures which is not the case for previous approaches
2. Improved performance over reasonable baselines
3. Method is relatively easy to understand


**Weaknesses**

1. Though the authors claim that the inference and training time cost is better for hub-transfer over ensembling, I am not 100% convinced. Note that the hub-transfer approach introduces significant overhead in terms of shuffling models in and out of memory as data-points are adaptively assigned.  Also - with the exploration bonus (and also early in training), it stands to reason that a batch of data could activate all models (though each data-point activates an individual subset but the union could be the whole hub) which presents a computational / memory hurdle to handle efficiently. Can the authors provide more details about how the approaches were implemented and benchmarked for Table 5 ?
2. Method performance  (in the case of out-of-domain transfer)  is on par with the ensembling approach which may be simpler to implement
3.  Seems technically complex to implement - would involve swapping models in and out of memory unless one has access to a large memory budget.
4. No error bars on results even though experiments were run for 3 seeds



**Update after rebuttal**
Score updated from 4->6 after rebuttal discussion

---

> ### Author Response · Authors · 2022-08-02
> **Response to Reviewer 8MEA (Part 1)**
>
> We would like to sincerely thank Reviewer 8MEA for providing the detailed review and insightful suggestions. We have revised our paper and appendix accordingly.
>
> **Q1:** More details about the implementation and costs of the method.
>
> We give more detailed complexity analysis and clarification of the proposed method.
>
> - The implementation of the approach.
>
> For Hub-Pathway, we load all the models into the memory instead of switching the model in and out of the memory. For each data point, we compute the pathway weight with the pathway generator and then forward the data point through its top-k activated models. All these processes are executed in the memory without shuffling costs. So the computational costs of Hub-Pathway is about k=2 times of a single model, much smaller than the Ensemble method in $\underline{\text{Table 5 of the original submission}}$, or $\underline{\text{Table 2 of the revised appendix}}$.
>
> - The memory cost of storing multiple models is much smaller than forwarding data through models.
>
> To validate it, we conduct the experiment on the classification task in CIFAR with a batch-size of 12, in which we (1) load the pre-trained models in the memory, (2) forward the data through the pathway generator, (3) forward the data through pre-trained models. We report the increase in memory costs (in Megabytes) of each step in the table below (These results are also included in the $\underline{\text{Table 4 of the revised appendix}}$). Comparing results in Column 1, the additional memory cost of storing multiple pre-trained models than one single model is about 300 M, which is relatively smaller than those brought by forwarding data through models (Column 3). This additional cost of storing more models is acceptable in most cases.
>
> | Method | Load Model | Forward Generator | Forward Models |
> | ------ | ------ | ------ | ------ |
> | Single | 897 | / | +1008 |
> | Ensemble | 1179 | / | +5218 |
> | Hub-Pathway | 1203 | +274 | +2060 |
>
> - The memory costs of forwarding data through models is controlled by Hub-Pathway.
>
> From the Column 3 of the table above, the memory cost of Ensemble scales up with the model size, to 5 times of the single model. The cost of Hub-Pathway is controlled by TopK selection with the pathway activation mechanism, and it will not become larger with even more pre-trained models. The pathway generator is lightweight, so its memory cost is small. Although the whole hub may be activated from the dataset view, each data point only goes through k models activated by it, controlling the memory and computation costs to about k times (a small k can achieve good performance as shown in $\underline{\text{Figure 3(c) of the original submission}}$). Besides, as discussed above, the memory cost of storing these models is also relatively small. In all, Hub-Pathway is a method to enable the efficient usage of multiple pre-trained models.
>
> **Q2:** Method performance compared with the ensembling approach.
>
> The proposed Hub-Pathway framework outperforms the ensembling approach in almost all the tasks. As shown in $\underline{\text{Table 5 of the original submission}}$ or $\underline{\text{Table 2 of the revised appendix}}$, Ensemble outperforms the single model with 1.09%, which indicates that an improvement of 1.09% is non-negligible. Our method further outperforms the ensemble with 1.13%, which demonstrates that our method outperforms the ensemble with a non-negligible gap. Furthermore, our method is more efficient in computation for both training and inference. The pathway only activates a fixed number of models, which can scale up to the increasing size of the model hub while the ensemble method cannot.
>
> **Q3:** About swapping models.
>
> As discussed in $\underline{\text{Qestion 1}}$, the cost of storing the models in the memory is acceptable in most cases and the memory cost of forwarding data is controlled by the pathway mechanism. So only in extreme cases with a limited memory budget, we need model swapping to sacrifice time for space, while in most cases, we load the pre-trained models in the memory without the additional costs of swapping models, which is also easy to implement.
>
> **Q4:** Error bars on results.
>
> We followed the reported results in the Zoo-Tuning paper, so the error bars were not reported in the original submission. We have now followed the advice of the reviewers' and reported error bars of our experiments in the $\underline{\text{revised paper}}$. The variance is small compared to the improvement of the Hub-Pathway method.

---

> > ### Comment · Reviewer_8MEA · 2022-08-06
> > **Rebuttal Response**
> >
> > Thank you for the detailed rebuttal and the attempts to address my concerns.
> > A few more clarifying questions :
> > 1. It seems all these figures you are reporting are for **inference** time ? My question below still stands for training time (especially early in training when the distribution over networks will be roughly uniform)
> >
> > *it stands to reason that a batch of data could activate all models (though each data-point activates an individual subset but the union could be the whole hub) which presents a computational / memory hurdle to handle efficiently*
> >
> > 2.  Your response also assumes you have enough memory to fit all the models. What happens if you do not have enough memory to fit all the available models ? How does your method compare to the following ensembling variants :
> >       1. ensemble all models by joint fine-tuning ?
> >       2. ensemble all models at test time but fine-tune models independently at train time ?
> >
> > Note that empirically (b) has been found to lead to better performance than (a) [1] and so one would usually do (b)
> >
> > [1] https://www.microsoft.com/en-us/research/blog/three-mysteries-in-deep-learning-ensemble-knowledge-distillation-and-self-distillation/

---

> > > ### Author Response · Authors · 2022-08-07
> > > **Response to Reviewer 8MEA (Part 1)**
> > >
> > > Thank you for providing timely comments and giving us the chance to further clarify our methods and results.
> > >
> > > We report the performance and complexity of four different methods during the training and testing time to help clarify the following questions. The memory cost metrics include the additional memory costs of loading all models than one model (Additional Model Memory) and the total memory during training (Train Memory) and testing (Test Memory). The computational cost metrics include the training and testing speed. We explore four different methods, fine-tuning one single pre-trained model (Single), fine-tuning the ensemble of 5 pre-trained models jointly and testing with their ensemble (Ensemble-Joint), fine-tuning 5 pre-trained models independently and testing with their ensemble (Ensemble-Independently), and Hub-Pathway. We have also updated the results and discussion in $\underline{\text{Section A.3 of the revised appendix}}$.
> > >
> > > | Method | Acc (\%) $\uparrow$ | Additional Model Memory (M) $\downarrow$| Train Memory (M) $\downarrow$| Test Memory (M) $\downarrow$| Train Speed (iters/s) $\uparrow$| Test Speed (samples/s) $\uparrow$|
> > > | ------ | ------ | ------ | ------ |------ | ------ | ------ |
> > > | Single | 83.41 | / | 2149 | 1905 | 10.87 | 484.92
> > > | Ensemble-Joint | 83.87 | 282 | 7497 | 6397 | 2.32 | 98.64
> > > | Ensemble-Independent | 84.50 | / | 2149 | 6397 | 10.87 / 5 = 2.17 | 98.64
> > > | Hub-Pathway | 85.63 | 306 | 4219 | 3537 | 4.68 | 240.48
> > >
> > > **Q1:** The computational and memory costs of different methods during the training time.
> > >
> > > From the table above, we can find that Hub-Pathway clearly introduces fewer computational and memory costs than Ensemble, with the costs being controlled at about 2 times compared with fine-tuning from one single model, while the costs of the ensemble methods scale up with the number of pre-trained models. Note that although Ensemble-Independent saves the training memory by fine-tuning one model each time, the total computational costs increase. Ensemble-Independent even takes longer time for training than Ensemble-Joint since the data processing and augmentation steps should be repeated for each model. It is true that all models can be activated, and thus we need to load all the models into the memory. However, from the table above, we can find that the additional memory costs of loading all models are relatively small compared with the total training memory costs. The main computational and memory costs come from forwarding data through the models, which are controlled by Hub-Pathway with the mechanism of data-dependent pathways to only activate top-k models for each data point (From the results in our paper, we find a small k can achieve good performance). In all, the additional computational and memory costs of training Hub-Pathway do not scale with the size of the model hub, which is also controlled and acceptable.
> > >
> > > **Q2:** What if we do not have enough memory to fit all models?
> > >
> > > The memory cost mainly consists of two parts: (1) storing the models in the memory and (2) forwarding data through the models to get outputs. As we discussed in $\underline{\text{Question 1}}$, the additional memory cost of using multiple models mainly lies in the latter one, which can be controlled by the top-k pathway activation mechanism of our method. If we do not have enough memory, we can choose a smaller k for each data point. As shown in $\underline{\text{Figure 3(c) of the paper}}$, although sometimes activating all models may get the best performance, we can already get good performance with a small number of k, which balances the efficiency and accuracy. Although Ensemble-Independent may reduce the training memory cost by training one model each time, Hub-Pathway is more memory-efficient during inference. And in most cases, the training environment may have more resources than the inference environment. Also, as shown in the table above, the additional memory cost of Hub-Pathway with a small k is not much, which is acceptable in most cases.

---

> > > > ### Comment · Reviewer_8MEA · 2022-08-07
> > > > **Response**
> > > >
> > > > Thanks again for your response :
> > > > I still have an issue with this statement -
> > > >
> > > > **As we discussed in  Question 1 , the additional memory cost of using multiple models mainly lies in the latter one, which can be controlled by the top-k pathway activation mechanism of our method. If we do not have enough memory, we can choose a smaller k for each data point.**
> > > >
> > > > Can you clarify the following for me :
> > > > 1. Assume k = 1. And assume that we have a batch size of data > the number of models so say bsz=32 and we have 5 models.
> > > > Early in training, for the 32 samples in the batch size - it is very unlikely that those 32 samples will activate **the same model**. They will each activate a single model, but taken together would activate all **5 models**. What happens in this case ?
> > > > Would you not have to either :
> > > >
> > > > a) swap models in and out of training
> > > >
> > > > b) for that batch, only update a subset of the models on each backward and forward pass - which means that you would be using a smaller effective batch size - and thus increasing your training time cost ?
> > > >
> > > > I think understanding this, and addressing this in the paper as a possible limitation, would strengthen the work.

---

> > > > > ### Author Response · Authors · 2022-08-08
> > > > > **Response to Reviewer 8MEA**
> > > > >
> > > > > Thanks for your timely comments again.
> > > > >
> > > > > In such cases, when data activate different models and the whole model hub are activated by the batch, we load all models in the memory, forward the data to the specific models activated by them, and train the activated models together.
> > > > >
> > > > > | Method | Total | Load Models |  Generator Output |  Model Output |
> > > > > | ------ | ------ | ------ | ------ | ------ |
> > > > > | Single | 4217 | 897 | / | +2692 |
> > > > > | Hub-Pathway | 5085 | 1203 | +598 | +2694 | 5085 |
> > > > >
> > > > > With batch-size=32, $k$=1, we report the memory cost (M) during training, where each data point activates one model, and all models are activated in the batch. We report the total memory cost and the memory cost after loading the model parameters. We also report the increment in memory costs after forwarding data to the pathway generator and pre-trained models to get output tensors. We make the following clarifications:
> > > > >
> > > > > - The memory costs are mainly composed of two parts: (1) loading and storing models in the memory and (2) forwarding data to models to get output tensors.
> > > > > - (1) is determined by the number of activated models in the batch (which is 5 in this experiment) since we load all activated models in the memory to avoid swapping models.
> > > > > - (2) is controlled by the number of models each data point forwards, i.e., the top-k selection hyper-parameter $k$ (which is 1 in this experiment) since the tensors are only created when a data point is forwarded through a model.
> > > > >
> > > > > From the second column, we can find that the additional cost of (1) in Hub-Pathway is small (about 300 M) compared with the cost of (2) (about 2600 M). From the last column, we can find that with $k$=1, Hub-Pathway has a similar cost of (2) compared with one single model, which justifies our clarification above.
> > > > >
> > > > > This motivates us to spend a few more costs storing all the models to avoid the costs of swapping models in and out of the memory. Because of this, our method has a complexity of about $O(k)$, which can be selected and controlled, while Ensemble has a complexity of about $O(m)$ ($m$ is the size of the model hub), making our method more efficient than Ensemble.
> > > > >
> > > > > As you have pointed out, it is a limitation of our work that our method introduces more costs than fine-tuning from a single model. We mainly discuss the situation of common pre-trained models where the cost of storing model parameters is not high compared with output tensors. Inspired by your comments, there may also be cases with large pre-trained models, where the costs of model parameters are higher than the output tensors. In such situations, loading all models in the memory is prohibitive, and we could seek to swap models in the memory to sacrifice time for space. We have updated these discussions of limitations in $\underline{\text{Section C of the revised appendix}}$. Thanks again for helping and guiding us to clarify these. We think these limitations do not hurt the core contribution of our method since it is still efficient and effective in common cases.

---

> > > > > > ### Comment · Reviewer_8MEA · 2022-08-08
> > > > > > **Score updated**
> > > > > >
> > > > > > Thanks for the response and for editing the paper to address concerns.
> > > > > >
> > > > > > Score has been updated from 4->6

---

> > > > > > > ### Author Response · Authors · 2022-08-08
> > > > > > > **Thank you for your comments**
> > > > > > >
> > > > > > > We sincerely thank Reviewer 8MEA for providing detailed and insightful reviews, which helped us strengthen our paper by providing more analysis and clarifying the strengths and limitations. Also, many thanks for the timely responses and the careful judgment of our feedbacks.

---

> > > ### Author Response · Authors · 2022-08-07
> > > **Response to Reviewer 8MEA (Part 2)**
> > >
> > > Continue with Q2
> > >
> > > We admit that there are some extreme cases where the resource cannot afford to store or forward even one more model, in which case our method may not work. We thank the reviewer for pointing out this limitation, and we have added this discussion into the $\underline{\text{Section C of the revised appendix}}$. We think this may not hurt the core contribution of our method because each method may have a scope of its application. For example, there are works designing deeper and bigger models for better performance and works designing lightweight models to sacrifice some performance for efficiency, both of which have their own values. In model hub transfer learning, we also need to trade-off between performance and efficiency, and from our reported results, our proposed method achieves a good balance between them. Of course, inspired by your comments, an ideal solution may achieve performance gain with almost no efficiency losses, which is a challenging goal and encourages our future exploration of this problem.
> > >
> > >
> > > **Q3:** Comparison with two ensembling variants.
> > >
> > > We compare the performance and complexity of Hub-Pathway with fine-tuning from one single model and the two variants of Ensemble proposed by the reviewer in the table above. Hub-Pathway outperforms the two variants of Ensemble, which shows the effectiveness of the proposed method. The training and testing costs of Ensemble-Joint all grow with the number of the models, making it inefficient for the problem. Ensemble-Independent saves the training memory by fine-tuning one model each time, but the computational costs during training and the costs during inference cannot be saved. Ensemble-Independent also takes a longer time for training than Ensemble-Joint, since some additional processes such as data processing and augmentation should be repeated for each model. Hub-Pathway also introduces additional costs than the single model baseline, but the additional costs can be controlled by pathway activation and thus do not scale up with the size of the hub. It also outperforms the two variants of Ensemble on most of the complexity metrics reported above. In all, Hub-Pathway achieves a better balance between the performance gain and the efficiency for the model hub transfer learning problem.

---

> ### Author Response · Authors · 2022-08-02
> **Response to Reviewer 8MEA (Part 2)**
>
> **Q5:** Are all the inputs similarly structured?
>
> All the models accept the same input structure, which means that all the pre-trained models should accept the same raw input, e.g., all models in Table 1 and 2 accept the image input. However, different pre-trained models can have different data preprocessing steps, which are included in the pre-trained models. For example, for a ResNet, we need to resize and crop the image, while for vision Transformers, we need to cut the image into patches.
>
> **Q6:** Why use a randomized sub-network.
>
> Temperature annealing can only control the smoothness of the pathway weight distribution but cannot create randomness to alternate the activated models. We need to explore the model hub and learn a picky pathway weight to activate the most suitable model, and the random subnetwork satisfies our requirement. Also, the random subnetwork is lightweight without much extra cost.
>
> **Q7:** The performance of a domain expert baseline.
>
> We explore a domain expert baseline, where for each task, we select the model with the best performance on this task. We have updated the tables to include the domain expert results in the $\underline{\text{revised paper}}$. The results of Table 1 are shown in the table below. We observe that the Hub_Pathway outperforms the domain expert baseline, indicating the importance of transferring knowledge from multiple models.
>
> | Method | CIFAR | COCO | Aircraft | Cars | Indoors | DMLab | EuroSAT | Avg. |
> | ------ | ------ | ------ | ------ | ------ | ------ | ------ | ------ | ------ |
> | Domian Expert | 81.18 | 81.97 | 84.97 | 89.38 | 73.69 | 75.06 | 98.82 | 83.58 |
> | Hub-Pathway | 83.31 | 84.36 | 87.52 | 91.72 | 76.91 | 76.47 | 99.12 | 85.63 |
>
> **Q8:** Comparison with the state-of-the-art methods on RL experiments.
>
> For the reinforcement learning experiments, we use the same model architectures for all the pre-trained models, which is exactly the problem setting of Zoo-Tuning, and thus, Zoo-Tuning performs well on this setting. Our method has a comparable performance with Zoo-Tuning and slightly outperforms it in the Gopher task. Compared with Zoo-Tuning, our method is more general to be applied to various heterogeneous architectures of pre-trained models. We have revised this claim in the $\underline{\text{revised paper}}$. We further add an experiments in $\underline{\text{Section A.6 of the revised appendix}}$. We increase the amount of the pre-trained models and use 5 models on 5 tasks: Seaquest, Riverraid, UpNDown, SpaceInvaders and BankHeist, and then transfer them to the Alien task. Results in $\underline{\text{Figure 2(c) of the revised appendix}}$ show that Hub-Pathway can still outperform Zoo-Tuning with the increased size of the model hub.
>
> **Q9:** The effect of the hyperparameters.
>
> For each dataset, we tune the hyper-parameters on one task and fix the hyper-parameters for other tasks. The hyper-parameters are not over-tuned and are efficient to select. We have already conducted an experiment on the influence of the hyper-parameter $k$ in $\underline{\text{Figure 3(c) of the original submission}}$. We add the experiment on the influence of $\lambda$ in the table below (also plotted in $\underline{\text{Figure 2(a) of the revised appendix}}$). We conduct the experiments on the COCO and Cars tasks for image classification. We observe that the performance is stable around $\lambda=0.3$. But without $\lambda$ ($\lambda=0$), the performance drops much, which indicates that the loss $L_\text{explore}$ is needed in our method to enhance the exploration of the model hub. .
>
> | $\lambda$ | 0.0 | 0.1 | 0.3 | 0.6 | 1.0 |
> | ------ | ------ | ------ | ------ | ------ | ------ |
> | COCO | 81.34 | 83.87 | 84.03 | 83.28 | 82.89 |
> | Cars | 89.75 | 92.34 | 91.90 | 91.39 | 91.16 |

---

### Official Review · Reviewer_6QEr · 2022-07-10

**Rating:** 5
**Confidence:** 4
**Soundness:** 3 good
**Presentation:** 3 good
**Contribution:** 3 good

**Summary:**

In this paper, the authors propose a transfer-learning method called Hub-Pathway. to leverage a library of pre-trained models instead of a single pre-trained model. The proposed Hub-Pathway utilized the idea of data-dependent transfer learning to find the best transfer learning path through multiple pre-trained models for each input. The output of each pre-trained model is then aggregated to produce the final prediction. The generator for generating the pathway is trained with softmax policy with additional noise to encourage exploration. The authors also propose exploitation loss to better leverage the activated model. Extensive experiments on multiple tasks demonstrate the benefits of the proposed approach.

**Questions:**

1. In Table 4, the ablation studies are only considered on COCO and Cars, I am wondering if the observation is consistent across all the test datasets.

2. What's the total inference time (pre-trained models + pathway generator) for a single image?

3. What's the effect of $\lambda$ and why $\lambda$ is 0.3?

4. Would the convergence of the model become slower because of the introduction of pathway finding?

**Limitations:**

Yes

**Strengths And Weaknesses:**

Strengths:

1. The paper is generally well-written and the organization is clear.

2. The idea of using multiple pre-trained models for transfer learning is interesting and the authors also propose a well-designed method for leveraging multiple models.

3. The experiments are extensive which show the effectiveness of the proposed method.


Weaknesses:
1. The additional storage is an overhead for using multiple pre-trained models for transfer learning. Also, the introduction of the pathway generator is another additional cost.

2. The key experimental details should be included in the main text rather than in the supplementary.  The choices of the hyperparameters are also not clearly specified.

3. Several related papers on dynamic computation and transfer learning are not cited.

4. Although the authors stated that the experiments are repeated for 3 times, no error bar is reported.

###########################

Post-rebuttal:

Thanks for the rebuttal. Most of my concerns are addressed in the rebuttal. I encouraged the authors to include the additional results in the final draft to improve the significance of the paper. I increase the score from 4 to 5.

---

> ### Author Response · Authors · 2022-08-02
> **Response to Reviewer 6QEr (Part 1)**
>
> Many Thanks to Reviewer 6QEr for providing the thorough insightful comments. We have revised our paper and appendix accordingly.
>
> **Q1:** The additional cost for using multiple pre-trained models and introducing the pathway generator.
>
> Recently, to improve model transferability, many works increase the model size and develop big models. We instead use the model hub to improve the model transferability. Both of these methods require larger storage and bring more costs for better performance. We want to clarify that even using multiple models, the additional costs can be controlled with properly designed transfer learning methods, and our method can achieve performance gains with acceptable costs.
>
> Firstly, we provided the complexity analysis in $\underline{\text{Table 5 of the original submission}}$, or $\underline{\text{Table 2 of the revised appendix}}$. It shows that:
> - The computation costs of using multiple pre-trained models can be controlled by Hub-Pathway, which do not scale with the size of the model hub.
> - The additional components of Hub-Pathway, including the pathway generator and aggregator, are lightweight and introduce only a few more parameters compared with the pre-trained models.
>
> Secondly, we delve into the question of storage cost or memory cost in this author response. We conduct the experiment on the classification task in CIFAR with a batch size of 12, in which we (1) load the pre-trained models in the memory, (2) forward the data through the pathway generator, (3) forward the data through pre-trained models. We report the increase in memory costs (in Megabytes) of each step in the table below (These results are also included in the $\underline{\text{Table 4 of the revised appendix}}$). We have the following observations:
> - The additional memory costs of storing the parameters of multiple models are relatively small compared with those brought by forwarding data through models. It is the latter one that damages the storage efficiency of using multiple models. [Explanation: Comparing results in Column 1, the additional memory cost of loading multiple pre-trained models is about 300 M, which is relatively smaller than those brought by forwarding data through models (Column 3).]
> - Hub-Pathway controls the memory costs of using multiple models. [Explanation: In Column 3, the memory cost of Ensemble scales up with the model size, to 5 times of the single model. The cost of Hub-Pathway is controlled by TopK selection with the pathway activation mechanism.]
> - The pathway generator is lightweight and brings few additional storage costs. [Explanation: In Column 1, the additional cost of storing the generator is negligible. Compared Column 2 with Column 3, the memory cost of forwarding through the generator is also relatively small compared with forwarding through the pre-trained models.]
>
>  In all, Hub-Pathway is a method to enable the efficient usage of multiple pre-trained models.
>
> | Method | Load Models | Forward Generator | Forward Models |
> | ------ | ------ | ------ | ------ |
> | Single | 897 | / | +1008 |
> | Ensemble | 1179 | / | +5218 |
> | Hub-Pathway | 1203 | +274 | +2060 |
>
>
> **Q2:** The experimental details.
>
> We clarify the key experimental details and the choices of the hyperparameters in $\underline{\text{Section 4 of the revised paper}}$.
>
> **Q3:** Some more related papers on dynamic computation and transfer learning.
>
> We have added some classic or recently published related works in the fields of dynamic computation and transfer learning in $\underline{\text{Section 2 of the revised paper}}$. Please point out other important works we missed during the discussion period.
>
> - Transfer learning:
>
> [1] Wortsman, Mitchell, et al. Model soups: averaging weights of multiple fine-tuned models improves accuracy without increasing inference time. In ICML, 2022.
>
> [2] Bolya, Daniel and Mittapalli, Rohit and Hoffman, Judy. Scalable Diverse Model Selection for Accessible Transfer Learning. In NeurIPS, 2021.
>
> [3] Huang, Long-Kai and Huang, Junzhou and Rong, Yu and Yang, Qiang and Wei, Ying. Frustratingly easy transferability estimation. In ICML, 2022.
>
> [4] Rebuffi, Sylvestre-Alvise and Bilen, Hakan and Vedaldi, Andrea. Learning multiple visual domains with residual adapters. In NeurIPS, 2017.
>
> [5] Aghajanyan, Armen and Zettlemoyer, Luke and Gupta, Sonal. Intrinsic dimensionality explains the effectiveness of language model fine-tuning. In ACL, 2021.

---

> ### Author Response · Authors · 2022-08-02
> **Response to Reviewer 6QEr (Part 2)**
>
> - Dynamic computation:
>
> [1] Brandon Yang, Gabriel Bender, Quoc V Le, and Jiquan Ngiam. Condconv: Conditionally parameterized convolutions for efﬁcient inference. In NeurIPS, 2019.
>
> [2] Yinpeng Chen, Xiyang Dai, Mengchen Liu, Dongdong Chen, Lu Yuan, and Zicheng Liu. Dynamic convolution: Attention over convolution kernels. In CVPR, 2020.
>
> [3] Robert A Jacobs, Michael I Jordan, Steven J Nowlan, and Geoffrey E Hinton. Adaptive mixtures of local experts. Neural computation, 1991.
>
> [4] William Fedus, Barret Zoph, and Noam Shazeer. Switch Transformers: Scaling to Trillion Parameter Models with Simple and Efﬁcient Sparsity. JMLR, 2022.
>
> [5] Han, Yizeng and Huang, Gao and Song, Shiji and Yang, Le and Wang, Honghui and Wang, Yulin. Dynamic neural networks: A survey, TPAMI, 2021.
>
> **Q4:** Error bars of the experiments.
>
> We followed the reported results in the Zoo-Tuning paper, so the error bars were not reported in the original submission. We have now followed the advice of the reviewers and reported error bars of our experiments in the $\underline{\text{revised paper}}$. The variance is small compared to the improvement of the Hub-Pathway method.
>
> **Q5:** Ablation studies across all test datasets.
>
> We conduct the ablation study of Hub-Pathway across all 7 tasks and report the results in the table below. The observation is consistent across these datasets. The pathway mechanism and the explore and exploit losses all help Hub-Pathway to better utilize the knowledge in the model hub.
>
> | Method | CIFAR | COCO | Aircraft | Cars | Indoors | DMLab | EuroSAT |
> | ------ | ------ | ------ | ------ | ------ | ------ | ------ | ------ |
> | w/ random path | 79.15 | 80.97 | 86.14 | 90.61 | 72.98 | 75.07 | 98.63 |
> | w/o explore | 80.02 | 81.34 | 85.72 | 89.75 | 73.06 | 75.78 | 98.59 |
> | w/o exploit | 78.32 | 80.10 | 86.38 | 90.59 | 72.02 | 75.95 | 98.65 |
> | Hub-Pathway | 83.31 | 84.36 | 87.52 | 91.72 | 76.91 | 76.47 | 99.12 |
>
> **Q6:** The total inference time for a single model.
>
> In $\underline{\text{Table 5 of the original submission}}$, or $\underline{\text{Table 2 of the revised appendix}}$, we reported the inference time of Hub-Pathway when it is tested with batches of data with the batch-size of 12. In this author response, we give a more detailed analysis of the inference time of a single image. We report the time (seconds) of forwarding a single image through the pathway generator and the pre-trained models in the table below (These results are also included in the $\underline{\text{Table 3 of the revised appendix}}$). The inference time of Hub-Pathway is about 2 times of a single model, much shorter than Ensemble, because of the pathway activation mechanism. The inference time of the generator is also small, compared with pre-trained models.
>
> | Method | Generator | Pre-trained Models |
> | ------ | ------ | ------ |
> | Single | / | 0.012 |
> | Ensemble | / | 0.056 |
> | Hub-Pathway | 0.003 | 0.023 |
>
> **Q7:** The effect of $\lambda$ and how it is chosen.
>
> $\lambda$ controls the trade-off between $L_\text{task}$ and $L_\text{explore}$. $\lambda$ is selected by the performance on one task in each dataset and we fix the chosen hyper-parameter for other tasks. We conduct an experiment on the influence of different $\lambda$ on the performance and report the results in the table below (also plotted in $\underline{\text{Figure 2(a) of the revised appendix}}$). The experiments are conducted on the COCO and Cars tasks for image classification. We observe that the performance is stable around $\lambda=0.3$. But without $\lambda$ ($\lambda=0$), the performance drops much, which indicates that the loss $L_\text{explore}$ is needed in our method to enhance the exploration of the model hub.
>
> | $\lambda$ | 0.0 | 0.1 | 0.3 | 0.6 | 1.0 |
> | ------ | ------ | ------ | ------ | ------ | ------ |
> | COCO | 81.34 | 83.87 | 84.03 | 83.28 | 82.89 |
> | Cars | 89.75 | 92.34 | 91.90 | 91.39 | 91.16 |
>
> **Q8:** The convergence of the model.
>
>  We plot the training curve for the Cars task in the image classification dataset in $\underline{\text{Figure 2(b) of the revised appendix}}$. We compare the convergence speed of Hub-Pathway and a single ImageNet model. We observe that in the first stage Hub-Pathway converges a little slower because of pathway finding, but then it converges with a similar speed to the single ImageNet model.

---

> ### Author Response · Authors · 2022-08-08
> **Response to Reviewer 6QEr**
>
> Dear Reviewer 6QEr,
>
> Thank you very much for your time and efforts in reviewing our paper.
>
> We want to kindly remind you that the author-reviewer discussion period will end in a few days. We have made an extensive effort to try to successfully address your concerns, by clarifying the complexity of our method, providing more analysis and experimental results, and making revisions to the paper and appendix.
>
> If you have any further concerns or questions, please do not hesitate to let us know, and we will respond to them timely.
>
> All the best,
>
> Authors

---

> > ### Author Response · Authors · 2022-08-09
> > **Thank you for your comments**
> >
> > Many thanks for your efforts in reviewing our paper and responses, and your comments to help improve our paper. We add the results in the rebuttal revision and they will be included in our paper.

---

### Official Review · Reviewer_5zEn · 2022-07-12

**Rating:** 6
**Confidence:** 3
**Soundness:** 3 good
**Presentation:** 4 excellent
**Contribution:** 3 good

**Summary:**

The paper proposes a method to transfer knowledge from multiple diverse models. First, it learns pathway routes that activate a different set of models depending on input data. Later, it aggregates the outputs to generate task-specific predictions. The authors propose an exploration loss to promote the utilization of all the models. Finally, the paper discusses the efficient fine-tuning of models on a subset of data that activates them. The paper claims state-of-the-art performance for model hub transfer learning on computer vision and reinforcement learning tasks.

**Questions:**

- Please refer to the weaknesses section. A bit more clarification on those aspects would be appreciated.
- How Hub-Pathway fares against data drift compared to pre-trained models? Likely, some pre-trained models are more robust against data drift; however, Hub-Pathway fails to activate them?

**Limitations:**

There could be limitations on what all heterogeneous model architectures Hub-Pathway could support. Please refer to the weaknesses section.

**Strengths And Weaknesses:**

Strengths:
- The paper is well-organized and clearly written.
- The paper's primary idea to facilitate knowledge transfer from multiple pre-trained models is important.
- The paper proposes a novel approach for transfer learning from multiple models. Hub-Pathway learns to identify top-k models best suited for input datum. The aggregator network generates a final prediction based on the output of k models and path weights.
- The authors propose an interesting exploration strategy to improve the transferability and exploitation of activated models via selective fine-tuning.
- The paper highlights the high efficiency compared to ensembles since not all models are active for both training/inference phases.

Weaknesses:
- The paper makes a broad claim on the effectiveness of the proposed method on heterogeneous model architectures. However, it's unclear how it could be used for autoregressive models where subsequent output tokes depend on the previous ones.
- The paper discusses replacing output head layers with task-specific layers (#119-#120). It is unclear how aggregator network could be set up for heterogeneous models with different output dimensions.

---

> ### Author Response · Authors · 2022-08-02
> **Response to Reviewer 5zEn**
>
> We would like to sincerely thank Reviewer 5zEn for providing the insightful review.
>
> **Q1:** How could Hub-Pathway be used for autoregressive models?
>
> The main difference between autoregressive models and those explored in our experiments is that autoregressive models process a sequence of inputs and output a sequence recursively. This has expanded the design space of the Hub-Pathway framework, and it has the potential to be used for such models with properly designed pathway generators and aggregators.
>
> - Pathway generator: A simple implementation is to flatten and concatenate the input tokens as the inputs of the generator. It may be a better choice to employ a model with the ability of sequence modeling, like LSTMs or Transformers, as the pathway generator.
> - Aggregator: A direct choice is to use the vanilla aggregator, which waits for each autoregressive model's prediction $Y_i=\Theta_i(X)$, where $Y_i=\{y_i^1,y_i^2,\cdots,y_i^{n}\}$, with $y_i^k=\Theta_i(y_i^{\leq k-1},X)$, and aggregates all these output sequences in a sequence level: $Y=A(Y_1,Y_2,\cdots,Y_m)$. For some autoregressive tasks with dynamic output lengths, we can also use such aggregators by padding the outputs of all models to fixed lengths. There may be other more complicated design choices. For example, it would also be possible to implement a recursive aggregator, which aggregate the outputs step by step: $y^k=A(y^{\leq k-1}, y_1^{k},y_2^{k},\cdots,y_m^k)$.
>
> Thanks for pointing out this problem, and we think it would be an interesting topic to explore deeper in model hub transfer learning problems on the autoregressive tasks in the future work.
>
>
> **Q2:** How could the aggregator be set up for heterogeneous models?
>
> For each pre-trained model, we remove the original task head and add a new target task head to map the original features to the target task output space. Because the models are utilized to solve the same downstream task together, their transformed outputs, which are the inputs of the aggregator, have the same fixed dimension. The aggregator then concatenates and aggregates these transformed outputs, so we just need the input dimensions of the aggregator to correspond to the transformed output dimensions. Thus, the different dimensions of heterogeneous models' original outputs would not be a problem for Hub-Pathway.
>
>
> **Q3:** The robustness against data drift.
>
> The pathway weights are fine-tuned on the target task, which activates the models that are most suitable for the target data. So these activated models are robust to the data drift between the source and target domains. As the results in $\underline{\text{Figure 3(a)}}$, the ImageNet model, which is commonly considered as a general model for image classification, has advantages in the pathway weights, which shows that Hub-Pathway has the ability to find more robust models. This question introduces the topic of obtaining transferred models from the model hub that can also generalize to other similar tasks or domains, which is not the focus of our paper, where we focus on transferring to one target task. We think it is an exciting and valuable problem in the future work.

---

> > ### Comment · Reviewer_5zEn · 2022-08-09
> > **Rebuttal Response**
> >
> > Dear Authors,
> >
> > Thank you for responding to the questions. This work is really interesting, and I, therefore, stand by my (accepting) recommendation.

---

> > > ### Author Response · Authors · 2022-08-09
> > > **Response to Reviewer 5zEn**
> > >
> > > Thank you for providing inspiring comments and timely responses.

---

### Meta-Review · Area_Chair_dCGN · 2022-08-22

**Recommendation:** Accept
**Confidence:** Certain

**Metareview:**

The submission introduces an approach called Hub-Pathway to leverage a diverse collection of pre-trained models for transfer learning. Hub-Pathway trains a pathway generator network to route examples to various models in a data-dependent manner and aggregates the outputs to produce task-specific predictions. Noise is added to the pathway generator and its output is entropy-regularized to encourage exploration, and the activated models are also individually trained on the target loss to encourage exploitation. The approach is evaluated on several image classification, facial landmark detection, and reinforcement learning tasks.

Reviewers noted the paper's clarity and writing quality and found the empirical evaluation extensive and convincing. On the other hand, they expressed doubts regarding Hub-Pathway's computational and memory complexity at training and inference time, in particular in comparison to the alternative of using an ensemble of models.

The authors responded by citing existing results in the submission and providing new results in the revised appendix showing that Hub-Pathway does in fact have lower computational complexity than an ensemble. They also argued through additional results that the forward propagation through multiple models (as opposed to holding the parameters of multiple models in memory) is the main memory bottleneck (which model ensembles also face), and that Hub-Pathway does better in that regard than model ensembles due to the pathway activation mechanism.

Overall the authors' response was satisfying to the reviewers, and their consensus is that the submission should be accepted. I therefore recommend acceptance.

**Award:**

No

---

### Decision · Program_Chairs · 2022-09-14

Accept